# Quantifying frequency content in cross-sectional retinal scans of diabetics vs. controls

**Joel A. Papay** [ID] [symbol], **Ann E. Elsner** [ID] *[symbol]

School of Optometry, Indiana University, Bloomington, Indiana, United States of America

[symbol] These authors contributed equally to this work.
* aeelsner@indiana.edu

**Data Availability Statement:** Data are held in a public repository (Harvard Dataverse, https://doi.org/10.7910/DVN/GH5MLF).

**Funding:** This work was supported by grants from the National Institutes of Health National Eye

## Abstract

### Purpose

To examine subtle differences in the structure of diabetic vs. control retinas.

### Methods

Spectral-domain optical coherence tomography (SD-OCT) images were compared for the retinas of 33 diabetic subjects who did not have clinical evidence of diabetic macular edema and age-matched controls, with central macular thicknesses of 275 and 276 microns, respectively. Cross-sectional retinal images through the fovea, called B-scans, were analyzed for spatial frequency content. The B-scans were processed to remove and smooth the portions of the retinal image not within regions of interest in the retina. The remaining retinal images were then quantified using a Fast Fourier Transform (FFT) approach that provided amplitude as a function of spatial frequency.

### Results

The FFT analysis showed that diabetic retinas had spatial frequency content with significantly higher power compared to control retinas particularly for a deeper fundus layer at mid-range spatial frequencies, ranging from $p = 0.0030$ to $0.0497$ at 16.8 to 18.2 microns/cycle. There was lower power at higher spatial frequencies, ranging from $p = 0.0296$ and $0.0482$ at 27.4 and 29.0 microns/cycle. The range of mid-range frequencies corresponds to the sizes of small blood vessel abnormalities and hard exudates. Retinal thickness did not differ between the two groups.

### Conclusions

Diabetic retinas, although not thicker than controls, had subtle but quantifiable pattern changes in SD-OCT images particularly in deeper fundus layers. The size range and distribution of this pattern in diabetic eyes were consistent with small blood vessel abnormalities and leakage of lipid and fluid. Feature-based biomarkers may augment retinal thickness criteria for management of diabetic eye complications, and may detect early changes.

Institute EY007624 (AEE), EY020017 (MSM), EY024315 (SAB) and P30EY019008 (SAB) (https://www.nei.nih.gov/). The funders had no role in study design, data collection and analysis, decision to publish, or preparation of the manuscript.

**Competing interests:** The authors have declared that no competing interests exist.

## Introduction

Neural and vascular changes in the retina are a main feature of diabetic retinopathy and diabetic macular edema, the sight threatening complications in the retina of many diabetics [1]. The clinical classification of the stages of severity of disease includes a variety of vascular changes, including both changes to the blood vessels and the leakage of fluid and lipid [2, 3]. Most grading and classification schemes have not yet evolved to make the best use of modern imaging techniques and instead rely on color fundus photography, with the original cameras using flood illumination and lacking scanning or confocal apertures to increase contrast. This is especially important in diabetic eyes, which are known to have ocular media problems [4].

It has long been known that there are sight-threatening retinal complications that could not be detected with older imaging and clinical examination methods [4, 5]. It has been shown recently that changes to retinal vessels can be far more severe than the clinical classification, visualized by using adaptive optics scanning laser ophthalmoscopy to increase both magnification and contrast [4, 6, 7]. Both retinal vessels and leakage from blood vessels, including blood lipids and proteins, are detected in eyes thought to have only mild or moderate diabetic retinopathy. In addition to leakage from blood vessels, there may also be leakage due to the disruption of the retinal pigment epithelium (RPE), so that fluid from the choriocapillaris leaks into the outer retina but is not pumped out fast enough to prevent the buildup of fluid [8]. Even when there is not necessarily fluid buildup, photoreceptors are still likely disrupted or damaged. It takes more light to bleach diabetic cones than compared to controls, indicating that they are not capturing light as efficiently [5]. Some diabetic subjects develop regions where cones do not wave guide light properly, indicating that the cones are likely physiologically disturbed [9]. Cysts in the outer retina have been associated with having poorer visual acuity [10].

The measurement of retinal thickening provides a widely adopted method of detecting and managing diabetic retinopathy and diabetic macular edema, often comparing one or more metrics of a patient's retina to a normative database, and interpreting the data in terms of ruling out other reasons for the thickening [11]. The measurement of retinal thickness is typically performed with spectral domain optical coherence tomography (SD-OCT), with recent techniques providing clear-cut cross sections in which features can be assessed for tissue disruption and causes of retinal thickening that are not attributed to fluid leakage from the diabetes, such as cyst-like structures where there are no blood vessels or thickening or thinning of individual structures within the retina including shortening of photoreceptors [4, 12–16]. Examples of such features also include fluid leakage, but due to retinal traction when a detaching vitreous pulls on the retina, which is a common finding during aging (Fig 1). Additional causes are many, but include other conditions such as new vessel growth and/or the development of permanent retinal scar tissue in patients with age-related macular degeneration or high myopia, both the leading causes of visual impairment in older individuals worldwide [4, 17].

An increase in retinal thickness may not be a sensitive measure of retinal pathology for a number of reasons. A common pathological change in diabetic patients' retinas is the damage and loss of neural elements [1, 18]. Additional pathological changes, such as those leading to trans-synaptic degeneration of retinal ganglion cells that results in retinal thinning, can also be the result of other diseases, often found with increasing age, such as stroke [19, 20]. A patient could have dying neurons that lead to retinal thinning at the same time as fluid leakage that is leading to retinal thickening. Further, some retinal changes result in one layer being thinned while another is thickened [21]. Thus, feature-based classification schemes are being developed [12, 14–16]. At present, these depend upon trained human graders for assessment, which is time-consuming and subjective. With approximately 28 million people with diabetes estimated

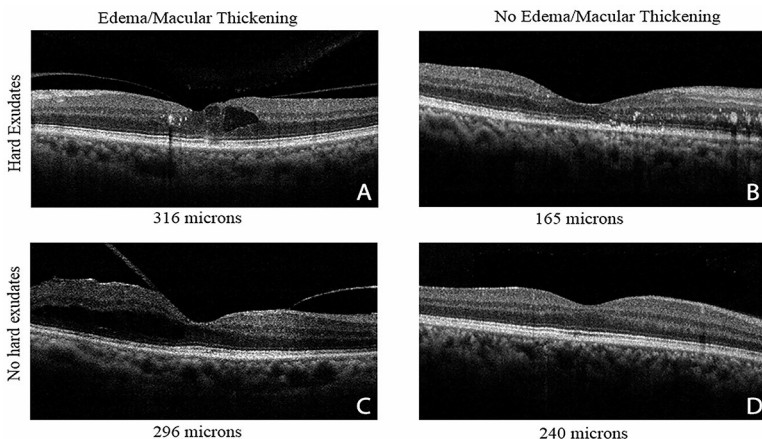

**Fig 1. Differences among diabetic eyes that are not fully characterized by retinal thickness, demonstrated with cross-sectional views and average central macular thickness from an iVue SD-OCT.** (A) Typical diabetic macular edema, showing a thickened retina that results in a high value of central macular thickness. There are fluid-filled cystic spaces that appear dark, hyper-reflective lipid and protein deposits (hard exudates), and disrupted photoreceptor layers that lie beneath retina blood vessels. The large areas of fluid produce multiply scattered light, instead of leading to interference, and therefore appear dark. (B) Significant pathological changes, but a low value of retinal thickness typically attributed due to damage to neurons and their support cells. There are numerous hard exudates, and significant disruption to retinal layers including photoreceptors. (C) Traction along with a detaching vitreous, the topmost reflective layer that is tilted, leading to retinal thickening, but not very severe diabetic changes. The large black areas within the retina demonstrate the fluid built up by the traction and are consistent with a high value of central macular thickness. (D) Diabetic retina with normal retinal thickness and minimal diabetic changes. These data are from the large dataset collected in a group of clinics for the underserved in Alameda County, CA, as described in ref [4].

to have vision threatening retinopathy worldwide, more efficient methods to detect retinal changes are needed [4, 16].

An alternative is to consider a combination of methods that provide thickness and features. At high magnification, we have shown that hard exudates can be seen in great numbers in eyes not classified within the more severe categories of diabetic retinopathy [6], and other researchers have noted other small and reflective structures [22]. The potential optical signatures that would indicate early changes to vessels or vessel leakage are often small features, e.g. small, hyper-reflective hard exudates or hyper-reflective structures within vessels. These features can be challenging to detect with clinical means or wide field imaging. Thus, we probed and quantified the differences between controls and diabetic patients in the spatial frequency content in OCT images, prior to retinal changes leading to increased or decreased retinal thickness in diabetic eyes.

## Methods

### Subjects

Subjects for the OCT computations were recruited from the Indiana University School of Optometry clinic. The diabetic subjects were diagnosed as not having diabetic retinopathy or macular edema during a comprehensive ophthalmological exam by a faculty member. The duration of diabetes was self-reported for all but one subject, ranging from 1–25 yr, mean = 6.78 ± 6.06 yr. The HbA1c was self-reported as < 7 by 18 of 33 subjects, as > 7 by 7 subjects, and unreported by 8 subjects. Thirty-three subjects with diabetes and 33 age- and sex-matched controls were recruited. There were 15 males and 18 females in each group. One of the diabetic subjects had Type 1 diabetes, with the rest having Type 2.

Written informed consent was obtained from all of the subjects, and the experiments conformed to the principles expressed in the Declaration of Helsinki. This research was approved by the Indiana University Institutional Review Board for all subjects. We selected subjects to illustrate the problem with limiting the diagnosis of pathological changes in diabetic patients to only central macular thickness values, shown in Fig 1, from a diabetic retinopathy screening study with consent and study approval also through the University of California Berkeley and Alameda Health for the subjects.

There was no statistical difference in age between the diabetics and controls, with the diabetics having a mean age of 58.9 yr and the controls having a mean age of 58.1 yr (p = 0.703). There was no statistical difference between males and females for age (p = 0.413), with males having a mean age of 57.5 yr and females having a mean age of 59.4 yr. There was also no statistical difference in the refractive errors between the groups (p = 0.220). The diabetics had a mean spherical equivalent error of -0.86D, with a standard deviation of 1.70D, and had a range from -4.25D to +2.00D. Control subjects had a mean spherical equivalent error of -0.36D, with a standard deviation of 1.58D, and ranged from -4.50D to +1.86D. As most of the subjects were not highly myopic, alterations in retinal layers could not be attributed to myopic degeneration or errors in the samples between groups due to magnification.

## Instrumentation and imaging

Subjects were imaged within one year of exam using spectral domain optical coherence tomography (SD-OCT) (Spectralis, Heidelberg Engineering, Heidelberg, Germany). The data were analyzed in several ways. To investigate differences in retinal thickness on a coarse scale between diabetics and controls, retinal thickness was obtained for each region of the ETDRS grids, as given by vendor software. For 12 of the 132 (9%) ETDRS outer grid values data were unavailable, and analysis proceeded with these data missing rather than using imputation.

To examine differences on a finer scale than the whole thickness of the retina, individual retinal layers were quantified. The horizontal, foveal centered B-scan was selected for each subject, from a volume scan that was collected from the subject, with measurements from a 15 deg region. Images were exported using the option that allows each pixel to have the same axial and lateral resolution. The B-scans were automatically segmented using custom MATLAB software (Mathworks, Natick, MA), with the segmentation reviewed and corrected manually if necessary. The retinal segmentation algorithm was based on a method previously used for OCT images from patients with retinal disease [23]. Briefly, this algorithm searches for dark-to-light and light-to-dark boundaries within an image. Since the human retina is very orderly and well-organized when the tissue is healthy, and the borders between layers are salient features while within layer features are subtle in a healthy retina, the assumptions are warranted. The algorithm iteratively searches for new boundaries, being constrained by where the boundaries were already drawn. This segmentation algorithm was developed before the Spectralis had commercially available software to segment individual layers. Further, this software offers ease of manual correction, and has already been integrated with other software in use in the lab.

Retinal thicknesses for five domains (Fig 2) were computed from the segmented data of each B-scan, sampled at 1 deg intervals, from 7 deg nasal to 7 deg temporal to the fovea. These domains were the full retina (Fig 2A), the domain from the junction of the inner segments and the outer segments (ISOS) to the inner limiting membrane (ILM) (Fig 2B), the ISOS to the boundary between the inner plexiform layer (IPL) and the inner nuclear layer (INL) (Fig 2C), the boundary between the ISOS to the boundary between the nerve fiber layer (NFL) and the ganglion cell layer (GCL) (Fig 2D), and the boundary between the retinal pigment epithelium (RPE) and the choroid (CH) to the ISOS (Fig 2E).

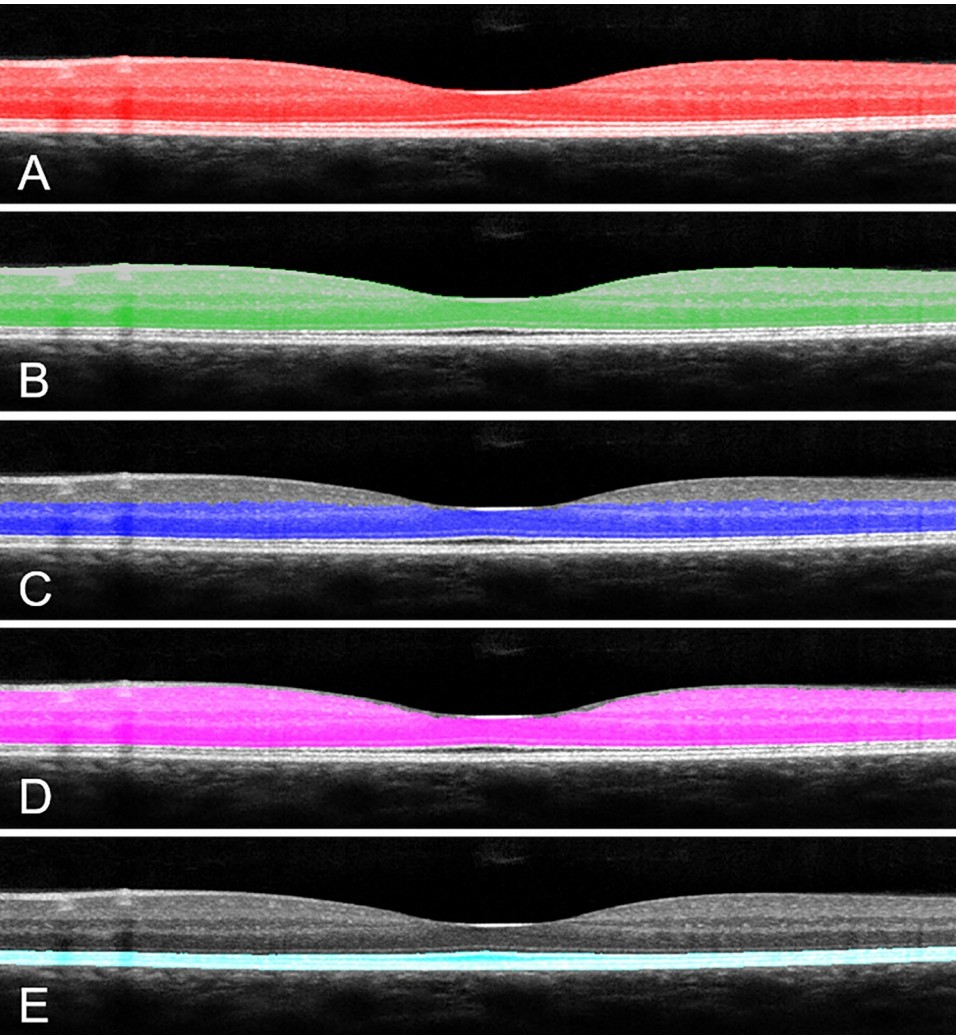

**Fig 2. The five domains where thickness and frequency content were measured.** A) The full retinal domain. B) The domain from the ISOS junction to the ILM. C) The domain from the ISOS to the boundary between the IPL and INL. D) The domain from the ISOS junction to the boundary between the NFL and GCL. E) The domain from the boundary between the RPE and CH to the ISOS junction.

A third analysis was performed to investigate the spatial detail that was not limited to thickness changes, by performing 2-dimensional Fourier analyses on these same five domains. A separate image was created for each domain, and then processed to reduce spurious frequencies, as follows. The images, 768 pixels wide, and always having a smaller height, were placed into a new image of 800x800 pixels, to reduce the complexity of frequency computations. All areas outside of the domain were set to have the same intensity as the mean intensity of the domain. The domains were all then flattened with respect to the ISOS junction, to avoid frequencies that could be introduced due to retinal shape [24]. Finally, the left- and right-most pixels of the domain were ramped to the mean intensity level to minimize edge effects (Fig 3).

Power spectra from the Fourier transforms were computed to analyze frequency content. Frequency was computed by measuring the fiduciary marks given on the images, which was 33 pixels per 200 microns, both laterally and axially. The number of microns per cycle for DC

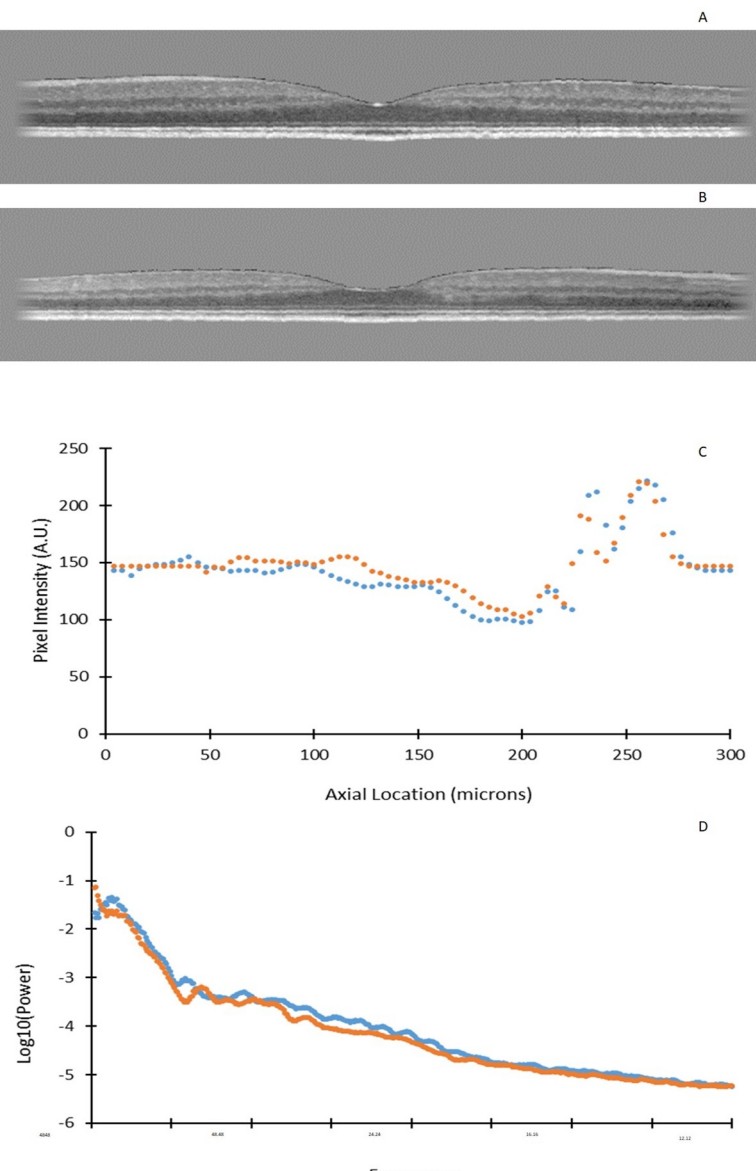

**Fig 3.** A) Representative b-scan from a diabetic subject, showing the full retinal domain, with the areas outside of the retina being set to the average intensity of the retina, and showing the ramping effects for edge smoothing on the sides of the retina. B) B-scan of the control subject. C) Average pixel intensity through the retina, with 0 as the inner-most position of the retina for the diabetic and control subjects in A and B, with the diabetic in blue and control in orange. D) Log transform of the power spectra for the subjects in A and B.

was computed as 4848.49, with the remaining microns per cycle being derived from that number. Paired one-tailed t-tests were used for comparison, because it was hypothesized that for the higher spatial frequencies, the diabetic subjects would have more power, due to small scale reflection or tissue property changes due to diabetes, such as to small blood vessels and other fine features. For lower spatial frequencies, the normal subjects were hypothesized to have more power due to the more regular layer structure, i.e. the grosser features. To examine the trends for individual subjects, z-scores of the amplitude distribution were computed at each frequency for each subject.

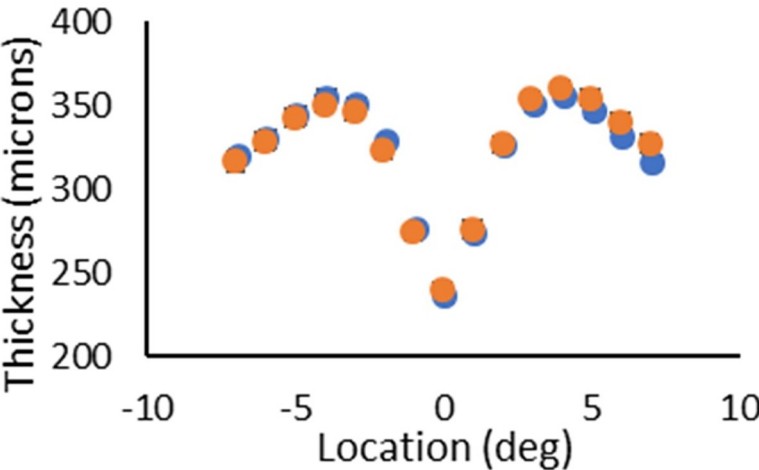

**Fig 4. Thickness measurements for the domain of the full retina.** Location 0 is the fovea, negative locations are in degree steps temporal to the fovea, and positive locations are in degree steps nasal to the fovea. There were no statistically significant differences in thickness between the diabetic and control subjects at any location, with diabetics in blue and controls in orange. That is, the trends seen in the figure are not significant, for diabetic subjects on average being thicker at all locations temporal to the fovea, and the control subjects being thicker on average at the fovea and for all locations nasal to the fovea.

## Results

### Retinal thickness results

There were no statistical differences in retinal thickness from the ETDRS regions between the groups using paired two-tailed t-tests (Fig 4). See Table 1. Additionally, there was no statistical difference in thickness for the central subfield for male vs female control subjects (p = 0.0794), having average thicknesses of 341 and 321 microns, respectively, although the trend is in the expected direction [25].

Thickness measurements of the domain from the boundary between the RPE and CH to the ISOS junction (Fig 5) were statistically significantly different only at two locations: 1 deg temporal to the fovea and 3 degrees nasal to the fovea (p = 0.0258 and 0.0116, respectively). At 1 degree temporal, the diabetics had an average thickness of 72.5 microns with a standard deviation of 4.8 microns, whereas the controls had an average thickness of 74.9 microns with a standard deviation of 3.6 microns. At 3 deg nasal, the diabetic subjects had an average thickness of 68.3 microns, with a standard deviation of 4.5 microns, whereas the control subjects

**Table 1. Retinal thicknesses and standard deviations for ETDRS regions.**

|  | Diabetics | | Controls | | |
| --- | --- | --- | --- | --- | --- |
| Region | Mean (μm) | St. Dev. (μm) | Mean (μm) | St. Dev. (μm) | p |
| Central | 275 | 20.2 | 276 | 26.1 | 0.800 |
| Inner Nasal | 343 | 19.1 | 344 | 21.3 | 0.854 |
| Outer Nasal | 310 | 19.0 | 316 | 25.9 | 0.339 |
| Inner Inferior | 335 | 19.4 | 340 | 19.8 | 0.227 |
| Outer Inferior | 291 | 18.7 | 301 | 19.2 | 0.083 |
| Inner Temporal | 326 | 17.8 | 330 | 20.6 | 0.409 |
| Outer Temporal | 282 | 17.2 | 288 | 17.9 | 0.212 |
| Inner Superior | 340 | 18.9 | 342 | 22.4 | 0.595 |
| Outer Superior | 302 | 17.9 | 304 | 22.4 | 0.383 |

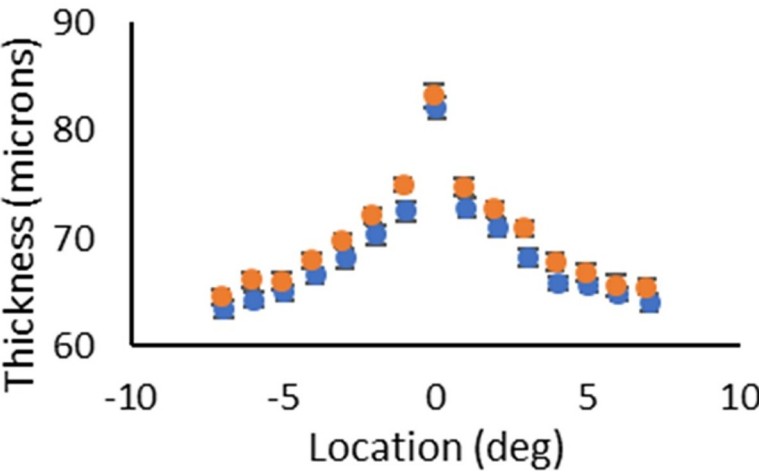

**Fig 5. Thickness measurements of the domain ranging from the boundary between the RPE and CH to the ISOS junction.** Location zero is the fovea, negative locations indicate degrees temporal to the fovea, and positive locations indicate degrees nasal to the fovea. Orange circles are the control subjects, and blue circles are the diabetic subjects. Control subjects were significantly thicker than the diabetic subjects at 1 deg temporal to the fovea and 3 deg nasal to the fovea. These data do not support the idea that diabetic subjects as a whole develop thicker deep retinal layers prior to clinical disease, for locations in the central retina.

had an average thickness of 71.0 microns, with a standard deviation of 4.1 microns. These results do not support early retinal thickening in the diabetic subjects for the deeper layers located in the central retina.

## Spatial pattern results measured with FFT

The distribution of power varied with frequency in a different manner for normal subjects compared with diabetic subjects, particularly for the deeper layers (Figs 2E, 6). The power differences were most notable in two frequency ranges (Fig 6A). The powers of the diabetics in the domain from the boundary between the RPE and CH to the ISOS junction were significantly less than those of the normal subjects in the frequency range of 25.9 to 29.0 microns/cycle (Fig 6B), with significance values ranging from p = 0.0296 at 27.4 microns per cycle to 0.0482 at 29.0 microns/cycle. In addition, the spectral power of the diabetics was significantly greater than that of the normal subjects in the frequency range of 15.5 to 18.2 microns/cycle (Fig 6C), with significance values ranging from p = 0.0030 at 16.8 microns/cycle to 0.0497 at 18.2 microns/cycle. The individual z-scores at each frequency (Fig 7) differed across groups for diabetic vs control subjects, with the diabetic subjects being above 0, i.e. more than average power, and many of the midrange frequencies. The diabetics had an overall average z-score 0.1665.

Spectral power of the diabetics was significantly greater than for the control subjects for the full retinal domain in the frequency range of 21.9 to 24.2 microns/cycle, with significance values ranging from p = 0.0245 at 22.9 microns / cycle to 0.0491 at 22 microns/cycle. Recall that there were no statistically significant differences in thickness for the 15 measured locations (Fig 4).

For the domain between the ISOS junction and the ILM (Fig 2B), there were three frequency ranges where there were significant differences between the diabetic subjects and the controls. The first frequency range corresponds to 210.8 to 346.3 microns/cycle, with significance values ranging from p = 0.0039 at 242 microns/cycle to 0.0429 at 346 microns/cycle. The second frequency range corresponds to 118 to 143 microns per cycle, with significance values

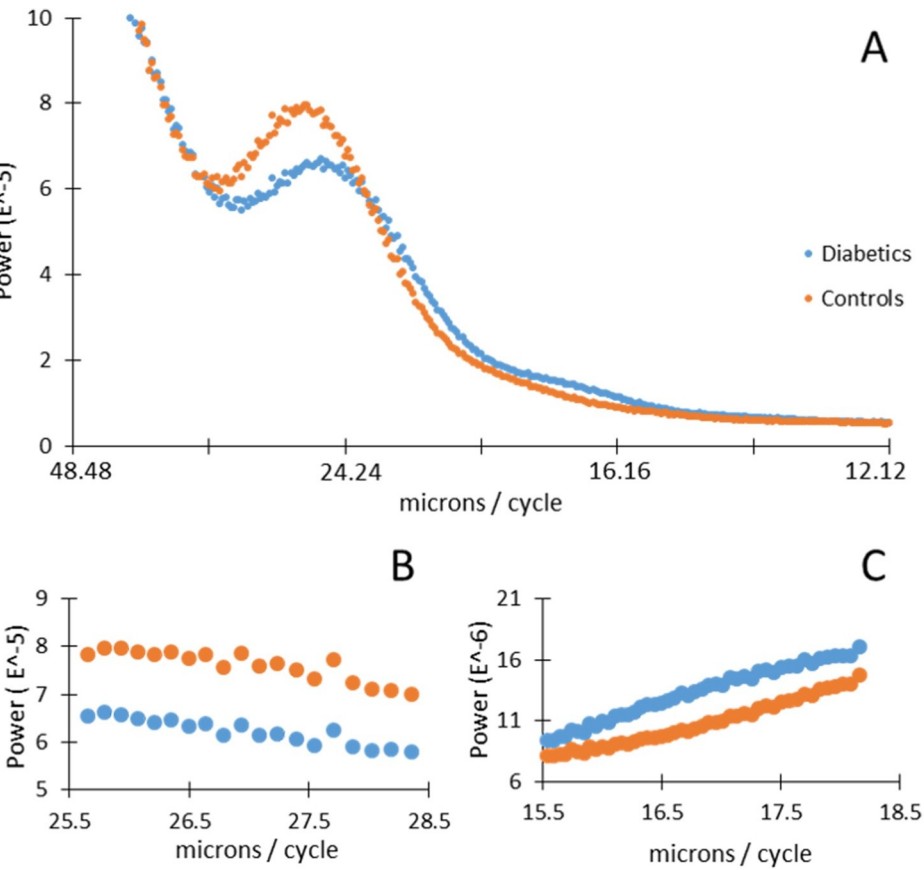

**Fig 6.** A) Average power spectrum for diabetics and controls for the domain that ranges from the boundary between the RPE and CH to the ISOS junction. B) Controls had statistically significant more power in the frequency range from 25.5 to 29 microns/cycle compared to diabetics. C) Diabetics had statistically significant more power in the frequency range from 15.5 to 18.2 microns/cycle compared to controls.

ranging from p = 0.0024 at 128 microns/cycle to 0.0492 at 143 microns/cycle. The third frequency range corresponds to 69.3 to 74.6 microns/cycle, with significance values ranging from 0.0129 at 72.4 microns/cycle to 0.0445 at 69.3 microns/cycle. For all three frequency ranges the diabetic subjects have the higher weighted powers.

The domain between the ISOS junction and the boundary between the NFL and GCL (Fig 2D) have similar results to the preceding domain. The diabetic subjects had higher weighted frequencies between 194 to 323 microns/cycle and from 110 to 128 microns/cycle. The first range had significance values ranging from p = 0.0072 at 220 microns/cycle and p = 0.0474 at 323 micron/cycle. The second range had significance values ranging from p = 0.0147 at 115 microns/cycle to 0.0434 at 110 microns/cycle. This close relationship was expected, due to the similarity between the domains.

The domain contained between the ISOS junction to the boundary between the IPL and INL (Fig 2C) had one frequency region where there were statistical differences between the diabetic and control subjects. The diabetic subjects had higher weighted frequencies from 138 to 211 microns/cycle. This region had significance values ranging from p = 0.0305 at 146.9 microns/cycle to 0.0467 at 211 microns/cycle.

In addition to frequency content, retinal thickness was also measured. In all ETDRS regions for the control subjects, a linear regression revealed that there was a decrease in thickness with

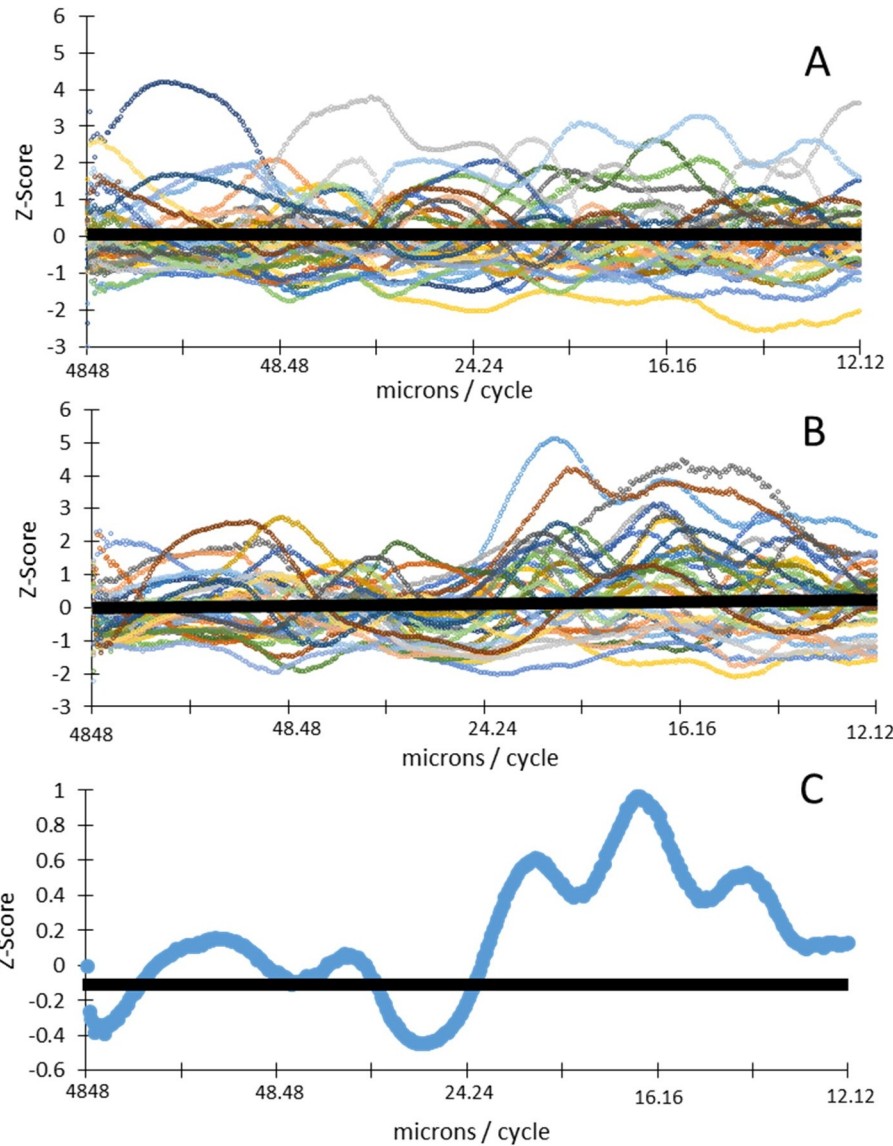

**Fig 7.** A) Individual z-scores for the control subjects computed for the power at each frequency for the domain from the boundary between the RPE and CH to the ISOS junction. B) Individual z-scores for the diabetic subjects in the same domain. C) The averaged z-score for the diabetic subjects at each frequency for the same domain. The diabetics consistently had more power at the higher frequencies than the control subjects.

increasing age. The association was weak however, with the largest r-squared value being 0.124, for the inner superior region. However, for the diabetic subjects, this was not the case. For the central subfield and the four inner regions, there was an increase in thickness as a function of age, but, like the control subjects, for the four outer regions, there was a decrease in thickness as a function of age. This association was also weak, with the largest r-squared value being 0.0333 (Table 2).

## Discussion

This study of 33 diabetic subjects without clinical signs of diabetic retinopathy or macular edema who were paired with control subjects analyzed the frequency content of OCT images

**Table 2. Linear regressions and R² values for age vs ETDRS region.**

| Region | Diabetics | | Controls | |
| --- | --- | --- | --- | --- |
| | Linear Regression | R² | Linear Regression | R² |
| Central | y = 0.217x + 262 | 0.0096 | y = -0.429x + 301 | 0.0247 |
| Inner Nasal | y = 0.127x + 335 | 0.0037 | y = -0.791x + 390 | 0.0037 |
| Outer Nasal | y = -0.0395x + 312 | 0.0004 | y = -0.761x + 359 | 0.0710 |
| Inner Inferior | y = 0.226x + 321 | 0.0112 | y = -0.570x + 373 | 0.0756 |
| Outer Inferior | y = -0.0207x + 293 | 0.0001 | y = -0.346x + 321 | 0.0277 |
| Inner Temporal | y = 0.357x + 305 | 0.0333 | y = -0.618x + 366 | 0.0822 |
| Outer Temporal | y = -0.0854x + 287 | 0.0020 | y = -0.241x + 301 | 0.0148 |
| Inner Superior | y = 0.156x + 330 | 0.0057 | y = -0.828x + 390 | 0.124 |
| Outer Superior | y = -0.226x + 315 | 0.0122 | y = -0.615x + 340 | 0.0122 |

as well as retinal thicknesses. Unlike previous studies that showed a significant difference in thickness at the foveal center of control eyes compared to diabetic eyes, even when there was no evidence of retinopathy in the diabetics [26], we found no statistical difference in thickness in our two groups. A thickness decrease is more likely detectable in the peripapillary region, as compared with the macular region [27]. Additionally, there was no difference in total retinal thickness for the nine ETDRS regions between the diabetics and the control groups. However, some statistically different thicknesses were found in the domain containing the RPE and OS. Control subjects had significantly thicker values than the diabetic subjects at 1 deg temporal to the fovea and 3 deg nasal to the fovea. Disruption to the outer layers of the retina in diabetics has been previously shown [27, 28].

Despite the lack of thickness differences between the control and diabetic subjects, the frequency analysis indicates that there are differences between these two groups, particularly for the domain contained between the RPE and CH boundary to the ISOS junction. This could be due to a buildup of lipids and proteins in the outer retina, potentially leading to the formation of hard exudates [28]. Molecular changes to vascular, neural, or glial tissue [1], including but not limited to vascular remodeling [6, 7], are also potential sources of the frequency content differences. Note that our findings are limited to the resolution of the instrumentation used. The frequencies with which the diabetic subjects had a higher weighted power is consistent with the sizes of small hyper-reflective foci and hard exudates. These changes cannot be solely attributed to the aging process, since both the diabetic and control patients had an increase of thinning of the retina in the four outer ETDRS regions with age, whereas the diabetic patients had an increase of thickness in the five inner regions and the control subjects did not. Further, there was no difference in age between the diabetic patients and the normal subjects.

This method for measuring the frequency content of retinal OCT images is objective and requires a trained grader only to ensure that the image segmentation is correct. Our data are consistent with detecting an enhancement of power with the spatial frequencies associated with the high frequencies of small structural changes, i.e. hyper-reflective foci and the precursors to hard exudates. There is also the decrease of power for the diabetic subjects for the lower frequencies in the domain of the deeper layers, consistent with less regularity of layer thicknesses or borders. The findings indicate that there is not a consistent trend for the diabetic subjects without diabetic retinopathy or macular edema to have increased thickness, and that the normal subjects sometimes have the thicker retinas. Thus, techniques with increased axial resolution will not improve the sensitivity of detection of the effects of diabetes on the retina for thickness measures in isolation of other information, since the diabetic retinas are not as a whole thicker and neural retinal thinning can occur early in the disease. Further, it is likely

that when only retinal thickness is used to classify patients, then false negatives can occur (Fig 2). The spatial frequency analysis is independent of retinal thickness, which did not differ in our sample because we selected diabetic patients with minimal retinal damage. These data are consistent with previous studies that have reported structural changes in diabetic retina that are in addition to, are at times independent from, retinal thickness [14, 15].

## Acknowledgments

We thank Drs. Jorge Cuadros, Glen Ozawa, and Taras Litvin for the collection of the diabetic data from the Alameda Health clinic for underserved patients. We thank Mr. Matthew Muller and Drs. Shane G. Brahm, Stuart B. Young, and Andréa V. Walker-Adeyemi for the organization of the data of the underserved patients, and Drs. Christopher A. Clark and Victor E. Malinovsky for the grading of the OCT images.

## Author Contributions

**Conceptualization:** Joel A. Papay, Ann E. Elsner.

**Data curation:** Joel A. Papay, Ann E. Elsner.

**Formal analysis:** Joel A. Papay, Ann E. Elsner.

**Funding acquisition:** Ann E. Elsner.

**Investigation:** Joel A. Papay, Ann E. Elsner.

**Methodology:** Joel A. Papay, Ann E. Elsner.

**Project administration:** Joel A. Papay, Ann E. Elsner.

**Resources:** Ann E. Elsner.

**Software:** Joel A. Papay, Ann E. Elsner.

**Supervision:** Ann E. Elsner.

**Validation:** Joel A. Papay, Ann E. Elsner.

**Visualization:** Joel A. Papay, Ann E. Elsner.

**Writing – original draft:** Joel A. Papay, Ann E. Elsner.

**Writing – review & editing:** Joel A. Papay, Ann E. Elsner.

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
