## [Decision Letter · Decision Letter 0]

30 Apr 2021

PONE-D-21-09236

Quantifying frequency content in cross-sectional retinal scans of diabetics vs. controls

PLOS ONE

Dear Dr. Elsner,

Thank you for submitting your manuscript to PLOS ONE. After careful consideration, we feel that it has merit but does not fully meet PLOS ONE’s publication criteria as it currently stands. Therefore, we invite you to submit a revised version of the manuscript that addresses the points raised during the review process.

Please address the concerns raised by the reviewer.

We look forward to receiving your revised manuscript.

Kind regards,

Tudor C Badea, M.D., M.A., Ph.D.

Academic Editor

PLOS ONE

Journal Requirements:

Reviewers' comments:

Reviewer's Responses to Questions

**Comments to the Author**

1. Is the manuscript technically sound, and do the data support the conclusions?

Reviewer #1: Yes

2. Has the statistical analysis been performed appropriately and rigorously? 

Reviewer #1: Yes

3. Have the authors made all data underlying the findings in their manuscript fully available?

Reviewer #1: No

4. Is the manuscript presented in an intelligible fashion and written in standard English?

Reviewer #1: Yes

5. Review Comments to the Author

Reviewer #1: The authors present a novel and potentially important analysis of SD-OCT images in persons with diabetes vs age-matched controls. The study is well-designed and presented. The key point is the difference in spatial frequency content between the two groups in a manner than varied with the spatial frequencies. Several points should be clarified:

1. The patients were not found to have visible microvascular lesions on clinical examination but it would be helpful to know how the exams were performed and whether or not they included fundus photographs.

2. It is not clear if the subjects were recruited only in Bloomington and/or in Alameda

3. Please provide the diabetes duration and hemoglobin A1c values if known for the diabetes subjects.

4. the cause of the differences cannot be determined from this study alone and subclinical vascular leakage is one possibility. Another could be molecular structural alterations from diabetes.

5. The authors modestly do not point out the innovation of the method but I think it is worthwhile if they wish to do so.

Hopefully the authors will perform longitudinal assessment of the subjects.

6. PLOS authors have the option to publish the peer review history of their article (what does this mean?). If published, this will include your full peer review and any attached files.

Reviewer #1: No

---

## [Author Response · Author response to Decision Letter 0]

25 May 2021

RE: PONE-D-21-09236

Quantifying frequency content in cross-sectional retinal scans of diabetics vs. controls

PLOS ONE

Thank you for the careful review of our article about quantifying the differences in frequency content from OCT images gathered from diabetic vs control subjects. We appreciate the amount of time it takes to appropriately review articles, and the suggestions that were made.

1. The patients were not found to have visible microvascular lesions on clinical examination but it would be helpful to know how the exams were performed and whether or not they included fundus photographs.

We have reorganized the wording of the Methods paragraph in response to items 1, 2, and 3:

Subjects for the OCT computations were recruited from the Indiana University School of Optometry clinic. The diabetic subjects were diagnosed as not having diabetic retinopathy or macular edema during a comprehensive ophthalmological exam by a faculty member. The duration of diabetes was self-reported for all but one subject, ranging from 1 – 25 yr, mean = 6.78 + 6.06 yr. The HbA1c was self-reported as < 7 by 18 of 33 subjects, as > 7 by 7 subjects, and unreported by 8 subjects. Thirty-three subjects with diabetes and 33 age- and sex-matched controls were recruited. There were 15 males and 18 females in each group. One of the diabetic subjects had Type 1 diabetes, with the rest having Type 2. 

 Written informed consent was obtained from all of the subjects, and the experiments conformed to the principles expressed in the Declaration of Helsinki. This research was approved by the Indiana University Institutional Review Board for all subjects. We selected subjects to illustrate the problem with limiting the diagnosis of pathological changes in diabetic patients to only central macular thickness values, shown in Fig 1, from a diabetic retinopathy screening study with consent and study approval also through the University of California Berkeley and Alameda Health for the subjects.

The patients in the OCT computation study were not patients from a diabetic retinopathy screening study, as were the patients in Fig. 1, but rather patients scheduled for their standard dilated fundus exams. Color fundus photographs are not routinely taken. 

On page 7, line 159, we added, ”Subjects were imaged within one year of exam using spectral domain optical coherence tomography (SD-OCT).” The effect of the potential development of retinopathy was minimal, since the OCT thickness values did not differ from control values. This is stated on lines 228-233.

2. It is not clear if the subjects were recruited only in Bloomington and/or in Alameda

We clarified that the OCT computation subjects were only from Bloomington by the above rewording in Methods. The Fig 1 subjects were from our screening study in Alameda, included to illustrate how diabetes can lead to not only thickening of the human retina, but also to thinning. The subjects recruited from Bloomington for this study do not exhibit those later stages of DR, as we want to detect earlier stages of the disease. Thus, the Bloomington subjects could not illustrate causes for abnormal central retinal thickness because as seen in Table 1, our diabetic and control subjects have similar retinal thickness based on the ETDRS grids from OCT.

3. Please provide the diabetes duration and hemoglobin A1c values if known for the diabetes subjects.

The duration data and HbA1c range are now provided where available in Methods, Lines 133-136. In our clinic, duration and HbA1c values are self-reported during routine clinical examinations, so they should not be used for scientific results. We are reporting what subjects state during their most recent clinical examination to the date of OCT testing.

4. The cause of the differences cannot be determined from this study alone and subclinical vascular leakage is one possibility. Another could be molecular structural alterations from diabetes.

We agree on this point. We changed the wording and recalled again references 1 (Antonetti) and 6 and 7 for vascular changes:

Lines 363-365 Molecular changes to vascular, neural, or glial tissue [1], including but not limited to vascular remodeling [6, 7], are also potential sources of the frequency content differences. Note that our findings are limited to the resolution of the instrumentation used. 

5. The authors modestly do not point out the innovation of the method but I think it is worthwhile if they wish to do so.

Thank you. We appreciate the comment and hope that this and other analyses expected from pathological changes will enter into classification methods, as this methodology can be implemented with basic software updates to analyze these kinds of data.

Hopefully the authors will perform longitudinal assessment of the subjects.

While only some of these subjects have test-retest data, the current grant in Dr. Burns’ lab, EY024315, is a longitudinal study. At each visit, the normal data collection includes OCT data of sufficient density and comparable macular grid size to permit further analysis.

Thanks for your time and consideration,

Dr. Ann E. Elsner and Joel A. Papay

---

## [Editor Report · Decision Letter 1]

28 May 2021

Quantifying frequency content in cross-sectional retinal scans of diabetics vs. controls

PONE-D-21-09236R1

Dear Dr. Elsner,

We’re pleased to inform you that your manuscript has been judged scientifically suitable for publication and will be formally accepted for publication once it meets all outstanding technical requirements.

Kind regards,

Tudor C Badea, M.D., M.A., Ph.D.

Academic Editor

PLOS ONE
---

## [Editor Report · Acceptance letter]

11 Jun 2021

PONE-D-21-09236R1 

Quantifying frequency content in cross-sectional retinal scans of diabetics vs. controls 

Dear Dr. Elsner:

I'm pleased to inform you that your manuscript has been deemed suitable for publication in PLOS ONE. Congratulations! Your manuscript is now with our production department. 

Kind regards, 

on behalf of

Dr. Tudor C Badea 

Academic Editor

PLOS ONE